# The Gut Microbiome among Postmenopausal Latvian Women in Relation to Dietary Habits

**DOI:** 10.3390/nu14173568

**Published:** 2022-08-30

**Authors:** Līva Aumeistere, Juris Ķibilds, Inese Siksna, Lolita Vija Neimane, Māra Kampara, Olga Ļubina, Inga Ciproviča

**Affiliations:** 1Faculty of Food Technology, Latvia University of Life Sciences and Technologies, Rīgas iela 22a, LV-3004 Jelgava, Latvia; 2Institute of Food Safety, Animal Health and Environment BIOR, Lejupes Street 3, LV-1076 Riga, Latvia; 3Faculty of Rehabilitation, Riga Stradiņš University, Dzirciema iela 16, LV-1007 Riga, Latvia; 4Children’s Clinical University Hospital, Vienibas gatve 46, LV-1004 Riga, Latvia

**Keywords:** gut microbiome, yoghurt, dietary habits, postmenopausal period

## Abstract

In recent years, many studies have been initiated to characterise the human gut microbiome in relation to different factors like age, lifestyle, and dietary habits. This study aimed to evaluate the impact of yoghurt intake on the gut microbiome among postmenopausal women and how overall dietary habits modulate the gut microbiome. In total, 52 participants were included in the study and two groups—a control (*n* = 26) and experimental group (*n* = 26)—were established. The study was eight weeks long. Both study groups were allowed to consume a self-selected diet, but the experimental group had to additionally consume 175 g of plain organic milk yoghurt on a daily basis for eight weeks. In addition, a series of questionnaires were completed, including a questionnaire on the subject’s sociodemographic background, health status, and lifestyle factors, as well as a food frequency questionnaire. Stool samples were collected for the analysis of the gut microbiome (both prior to and after the eight weeks of the study). Sequencing of V4-V5 regions of the 16S rRNA gene was used to determine the bacterial composition of stool samples. The dominant phylum from the gut microbiome was *Firmicutes* (~70% to 73%), followed by *Bacteroidota* (~20% to 23%). Although no significant changes in the gut microbiome were related to daily consumption of yoghurt, we report that consumption of food products like grains, grain-based products, milk and milk products, and beverages (tea, coffee) is associated with differences in the composition of the gut microbiome. Establishing nutritional strategies to shape the gut microbiome could contribute to improved health status in postmenopausal women, but further research is needed.

## 1. Introduction

The microbiota is all commensal microorganisms colonising the body (bacteria, archaea, viruses, and unicellular eukaryotes). The largest microbial population is found in the human gut with the total number of bacterial cells ranging from 10^13^ to 10^14^. The ensemble of genes contained by this microbiota is known as the “gut microbiome” [1,2,3,4]. Four major microbial phyla represent over 90% of gut microbiota—*Firmicutes*, *Bacteroidota*, *Proteobacteria*, and *Actinobacteriota* [5,6]. The phylum *Firmicutes* is composed of genus such as *Lactobacillus*, *Bacillus*, *Clostridium*, *Enterococcus*, and *Ruminococcus*. The phylum *Bacteroidota* consists of the genus such as *Bacteroides* and *Prevotella*. The phylum *Actinobacteriota* (main genus *Bifidobacterium*) and phylum *Proteobacteria* are less abundant [7].

The gut microbiome is a key contributor to many health aspects. For example, fermentation of a variety of indigestible dietary polysaccharides that further supports the growth of gut bacteria, which produce short-chain fatty acids such as butyric acid, propionic acid, etc. Butyric acid is the main energy source for the epithelial cells of the colon [1,4,6]. The gut microbiome is also a producer of vitamin K and B group vitamins, thus contributing to micronutrient sufficiency [8].

The composition of the adult gut microbiome is considered relatively stable. However, age and age-related states like menopause and postmenopause can lead to changes in the gut microbiome. For example, a lower abundance of phyla *Firmicutes* and *Roseburia* spp. are observed among postmenopausal women, while phylum *Bacteroidota* and genus *Tolumonas* are overrepresented [9]. In addition, age-related bone mineral density loss (osteopenia and osteoporosis) is more common among women due to changes in bone resorption and bone formation during menopause and the postmenopausal period [10]. One study [11] observed that the genus *Bacteroides* was more abundant among women with osteopenia and osteoporosis, but women with normal bone mass density had more unclassified *Clostridia* spp. and methanogenic archaea (*Methanobacteriaceae*) [11]. Other researchers [12] report that bacteria which represent the genus *Klebsiella*, *Escherichia, Shigella*, *Enterobacter*, *Citrobacter*, *Pseudomonas*, *Succinivibrio*, and *Desulfovibrio*, were more abundant among stool samples obtained from postmenopausal women with osteopenia but *Parabacteroides* spp., *Lactobacillus* spp. and *Bacteroides intestinalis* were more abundant in postmenopausal women with osteoporosis.

Other factors, like lifestyle (smoking, alcohol consumption, etc.) and dietary habits can also affect the composition of the gut microbiome [6,13]. For example, complex carbohydrate intake is associated with increased levels of *Bifidobacterium* spp., but intake of monounsaturated and omega-6 polyunsaturated fatty acids shows the opposite association [6]. Regular consumption of red meat increases the abundance of genus *Bacteroides*, but a higher total carotenoid intake (vegetable, fruit, berries) is associated with a higher overall gut microbiome diversity [6,14].

Fermented milk products contain live bacteria (mainly *Lactococcus spp*. and *Lactobacillus* spp.). One of the most popular fermented milk products around the world is yoghurt, with the highest consumption reported in Europe (accounting for as much as 32% of milk product intake in Europe) [15,16]. Yoghurt is produced using *Lactobacillus bulgaricus* and *Streptococcus thermophilus* and usually contains at least 10^7^ colony-forming units (CFU) per gram [17,18]. 

In recent years, many researchers have tried to evaluate if the consumption of yoghurt may modulate the gut microbiome supporting healthy microbiome development [1,18]. Previous studies have already reported that consumption of yoghurt and other fermented milk products can modify the gut microbiome by increasing the abundance of the genus *Lactobacillus* [1,5]. Another study showed that four weeks of probiotic yoghurt consumption by healthy adults increased the concentration of probiotics—*Lactobacillus acidophilus* LA-5 and *Bifidobacterium animalis* subsp. *lactis* BB-12—in the gut microbiome [19]. 

In Latvia, several studies [20,21] evaluating the gut microbiome among adults with different dietary habits have been started, but results are not yet summarised and publicly available. This study is unique as it aimed to not only evaluate how overall dietary habits modulate the gut microbiome in a specific target population but also how a specific product (yoghurt) included in the everyday diet can impact the gut microbiome. 

## 2. Materials and Methods

This study aimed to evaluate the impact of yoghurt intake on the gut microbiome among postmenopausal women in Latvia. Prior to the study, approval from the Riga Stradiņš University Ethics Committee was obtained (protocol code No. 22-2/278/2021. and date of approval—15 April 2021). Written informed consent was obtained from all participants before the study.

The inclusion criteria for participating women were:

45 to 69 years old;Postmenopausal (no menstrual bleeding for at least 12 months).

The exclusion criteria were:

3.Women receiving glucocorticoid therapy;4.Women diagnosed with osteoporosis prior to the study;5.Women with severe chronic diseases;6.Women with chronic digestive diseases and, therefore, following a specific diet.

In total, 52 women participated in this cross-sectional study from June 2021 to January 2022. Participants were recruited on the principle of convenience (from researchers’ own social networks). To evaluate the impact of yoghurt intake on the gut microbiome, participants were randomly divided into two groups—the experimental group with participants who were consuming 175 g of plain organic milk yoghurt daily for eight weeks (*n* = 26) and the control group (*n* = 26). An 8-week study period was selected based on similar research conducted previously (1,5,19). The list of ingredients, energy and nutritional value of the yoghurt (JSC Tukuma piens, Latvia) consumed among participants from the experimental group are listed in Table 1.

At the first meeting, the procedure of the study was explained to the participant, and a series of questionnaires were completed, including a questionnaire on the subject’s sociodemographic background, health status, etc., as well as a food frequency questionnaire. Participants’ bone mass density (evaluated at the femoral neck and lumbar spine region L1 to L4) was measured prior to the study using dual-energy X-ray absorptiometry (DEXA) by a radiologist trained in DEXA interpretation. Data regarding hip and waist circumferences were also collected using the anthropometric measurement protocol described by World Health Organization [22]. Hip and waist circumferences were taken using a measuring tape (Measuring tape 201, SECA, Hamburg, Germany). Data regarding weight and height were collected before the DEXA scan for bone mineral density. Based on obtained values, the body mass index was calculated.

The food frequency questionnaire consisted of 95 food products, beverages, and dietary supplements for which the frequency of consumption was estimated by the participants (participants had to mark how frequently specific products had been consumed in the last year). Food products and beverages from the food frequency questionnaire were divided into 21 categories, and average intake values for these categories were evaluated (Appendix A).

Stool samples were collected for the analysis of the gut microbiome (both prior to and after the eight weeks of the study). Until delivery to the laboratory, stool samples were stored in a refrigerator or other cold place (+2 to +4 °C). Samples were aliquoted and frozen at –80 °C until further processing. DNA was extracted using the ZymoBIOMICS 96 Magbead DNA Kit. DNA libraries for metataxonomic sequencing were prepared following the Microbiome Helper protocol [23]. Briefly, variable regions V4-V5 of bacterial 16S rRNA gene were amplified and indexed in a single-step PCR using KAPA HiFi DNA polymerase and 515FB/926R primers [24,25]. Libraries were sequenced on Illumina MiSeq, producing 2 × 300 bp paired-end reads.

Sequence reads were processed within the QIIME2 software environment [26]. Primer trimming with Cutadapt [27] was followed by denoising with the DADA2 algorithm [28], which produced error-corrected amplicon sequence variant (ASV) features. ASVs were taxonomically classified by VSEARCH [29] against the SILVA v138 NR99 SSU rRNA database [30]. Before further calculations, feature tables were filtered to remove singletons, unclassified and eukaryotic ASVs, and feature counts were normalised between samples using the SRS algorithm [31]. Alpha (within-sample) diversity was calculated as Shannon’s index [32]. Beta (between-sample) diversity was calculated as Bray–Curtis dissimilarity [33]. IBM SPSS Statistics 23 was used for food frequency questionnaires and other data statistical analysis. Non-parametrical statistical tests such as the independent samples Mann–Whitney U Test and independent samples median test were applied to analyse the data. To evaluate how overall dietary habits modulate the gut microbiome, non-parametric partial Spearman’s rank correlations were calculated while controlling for the following covariates (age, education level, body mass index, waist-to-hip-ratio, last menstrual period, smoking, use of alcohol, bone mineral density level, and use of antibiotics). A principal component analysis was conducted to identify different patterns related to dietary habits and gut microbiome composition among the participants. The count of patterns of principal component analysis was identified based on the eigenvalue (≥1). Values with the factor loading ≥|0.5| were considered to contribute significantly to the identified components. A *p*-value of ≤0.05 was considered statistically significant for all statistical tests. 

## 3. Results

### 3.1. Characteristics of the Participants Involved in the Study

The characteristics of the participants are compiled in Table 2. According to the body mass index, 17 of the participants had a normal body mass index (18.5 to 24.9 kg m^−2^), 18 participants were overweight (25.0 to 29.9 kg m^−2^), and 17 participants were obese (≥30 kg·m^−2^).

### 3.2. Dietary Habits among Study Participants

Data from the food frequency questionnaire are compiled in Appendix A. No significant differences were found regarding food, beverage, and dietary supplement intake among study groups, except for vitamin D dietary supplement intake, which was higher among the participants from the experimental group (average intake frequency “three to four times a week” reported in the experimental group and average intake frequency “two to three times a month” reported in the control group, *p* = 0.026) and coffee intake which was also higher among the experimental group (Appendix A, *p* = 0.039).

Participants who reported more frequent consumption of meat products were also consuming more frequently products like fast food, salty snacks (ρ = 0.700, *p* < 0.0001), and sweets and bakery goods (ρ = 0.648, *p* < 0.0001). There was a strong positive correlation between fast food, salty snacks and sweet and bakery good intake (ρ = 0.521, *p* = 0.002), and consumption of lemonades (ρ = 0.582, *p* < 0.0001). Meat intake correlated with potatoes intake (ρ = 0.581, *p* < 0.0001).

Older participants were consuming less vegetables (ρ = -0.289, *p* = 0.044), but participants with a higher waist circumference were consuming more meat products (ρ = 0.305, *p* = 0.033) and fish products (ρ = 0.362, *p* = 0.011). 

### 3.3. Description of the Gut Microbiome

The dominant phylum from the gut microbiome was *Firmicutes* (~70% to 73%), followed by phylum *Bacteroidota* (~20% to 23%). The rest of the gut microbiome consisted of phylum *Actinobacteriota* and other phyla (Figure 1).

No significant changes in the gut microbiome were related to the daily consumption of plain organic milk yoghurt for eight weeks in the experimental group (*p* > 0.05, Figure 2). No significant differences were found regarding the ratio of *Firmicutes*:*Bacteroidota*, the ratio of *Proteobacteria*:*Actinobacteria*, and alpha-diversity (Table 3, Figure 2) both within groups comparing samples before and after the study and between the control group and experimental group both before and after the study (*p* > 0.050 for all).

No significant differences were found regarding beta diversity comparing both study groups (*p* = 0.812) (Figure 2).

Age, time from the last menstrual period or other participants’ characteristics (described in Table 2) were not significant factors affecting the composition of the gut microbiome (*p* > 0.05).

Comparing data from the food frequency questionnaire and gut microbiome analysed before the study, the following observations were found–a significantly higher abundance of phylum *Actinobacteriota* were associated with fast food and salty snack intake. A higher vegetable intake was associated with a higher abundance of phylum *Bacteriodota* but a lower abundance of phylum *Bacteriodota* was associated with meat, offal, lemonade, and energy drink intake. Meat, offal, fast food, salty snack, lemonade, and energy drink intake was associated with a higher abundance of phylum *Firmicutes* (Figure 3).

Grains, grain-based product intake was associated with a higher abundance of genus *Bifidobacterium* and genus *Atopobium*. Potatoe intake negatively correlated with the abundance of *Enterobacter* spp. (Appendix A).

Meat and offal intake was negatively associated with the abundance of genus *Gastranaerophilales*, *Victivallis*, *Anaeroplasma*, *Terrisporobacter*, and *Enterobacter*. Fish product intake was associated with a higher abundance of order *Coriobacteriales*.

The abundance of *Subdoligranulum* spp. was associated with milk and milk product intake, but vegetable oil and plant- and animal-based fat intake with a higher abundance of genus *Campylobacter*, *Fusobacterium*, *Leptotrichia*, *Porhphyromonas*, *Faecalitalea*, *Gemelia*, *Epulopiscium*, *Hungatella*, *Parvimonas*, and family *Veillonellaceae*.

A higher vegetable intake was negatively associated with the abundance of family *Oscillospiraceae*, *Paludicola* spp., but a higher legume, nuts, seed, and milk alternatives intake with a higher abundance of *Citrobacter* spp.

Higher consumption of sweets and baked goods was negatively associated with the abundance of *Mitsuokella* spp., but a higher intake of condiments (sugar, honey, etc.) was associated with a higher abundance of *Anaerotruncus* spp.

A lower abundance of genus *Gastranaerophilales*, genus *Victivallis*, and genus *Anaeroplasma* were found among participants with a higher coffee intake, but a lower abundance of *Coprobacter* spp. was found among participants who reported a more frequent intake of tea. Family *Methanomethylophilaceae* was found less abundant among the participants who reported a more frequent intake of water.

Using principal component analysis, three dietary habits and gut microbiome composition-related profiles were identified, explaining 26.01% of the total profile variance (Appendix A):

the first profile was associated with a high abundance of phylum *Actinobacteriota*, low abundance of phylum *Bacteroidota*, and high intake of fast food and salty snacks, sweets, and bakery goods (explaining 11.71% of total variance);the second profile was associated with a high abundance of phylum *Bacteroidota*, low abundance of phylum *Firmicutes*, and high intake of vegetables, potatoes but low intake of legumes, nuts, seeds, and milk alternatives (explaining 8.81% of total variance);the third profile was associated with low abundance of phylum *Proteobacteria* and a low intake of lemonades and energy drinks (explaining 5.49% of total variance).

## 4. Discussion

National nutritional guidelines for adults [34] recommend consuming on a daily basis at least 500 g of vegetables, fruits and berries, of which at least 300 g should be vegetables and 200 g of fruits and berries. Unfortunately, the average vegetable intake among study participants was significantly lower. For only three participants (all from the experimental group), the daily vegetable intake reached the recommended 300 g per day. Overall, the calculated average daily intake of vegetables reached only ~125 to 180 g per day. Also, the average fruit and berries intake was slightly lower, reaching only around 160 to 170 g per day. Vegetables, fruits, and berries are important sources of vitamins, minerals, and antioxidants. Therefore, it is important to include them in an everyday diet regardless of age [35].

The daily intake of starchy food (grains, grain-based products, bread, potatoes) should reach 550 g [34], as they are an important source of energy, complex carbohydrates, vitamins, minerals, and fibre [35]. However, starchy food intake among study participants was very low—around 130 g per day. Grains and grain-based products provided the highest intake of starchy foods (~50%), followed by potatoes (~41%) and bread (~9%).

Daily milk and milk product intake should be two to three portions per day [34]. One portion, for example, is a glass (250 mL) of milk or fermented milk product (kefir, yoghurt, etc.), 100 g of cottage cheese, or 30 g of cheese [34]. The intake of milk and milk products among study participants was sufficient—around two portions per day as the average calculated daily intake of milk and fermented milk products reached approximately 230 to 260 g, the daily intake of cottage cheese—around 40 g, but the daily intake of cheese—30 g.

Legumes, meat, fish, and eggs as a protein, vitamins, and minerals sources should also be included in the everyday diet, preferably in two to three portions per day [34]. One portion is a glass (~150 g) of cooked legumes, 80 to 100 g of cooked meat, 100 to 140 g of cooked fish or 2 eggs (around 100 g) [34]. Overall, meat was the dominant protein source (average intake of 60 to 80 g or around one portion per day). Only around 35% of the participants in the food frequency noted the fish intake around one to two times a week, which is within national nutritional guidelines [34], and the average calculated daily fish intake reached only 30 g per day. Similar values were reported for daily egg intake (~28 g). Although legumes are healthy food products that provide nutrients like protein and fibre, their intake among study participants was reported as low (consumed mostly only a few times a year) and calculated daily intake reached only ~15 g per day.

Like previously reported information about the gut microbiome [7], the dominant phyla from the samples analysed in this study were *Firmicutes*, *Bacteroidota*, *Actinobacteriota*, and *Proteobacteria*, with the two phyla—*Firmicutes* and *Bacteroidota*—representing ~90% of the gut microbiome. While the overall gut microbiome of a healthy individual is relatively stable, its composition can be influenced by the person’s age, lifestyle (alcohol intake, smoking, etc.), and dietary habits [7].

Ageing has been associated with a decline in microbial diversity (a decrease in *Bifidobacterium* spp. and an increase in *Clostridium* and *Proteobacteria* spp.) and changes in the ratio of *Firmicutes*:*Bacteroidota* (ratio of 10.9 in adults, 25–45 years old, and ratio of 0.6 for 70 to 90 years old) [7,36]. This could be due to changes in the digestion of elderly people (for example, impaired intestinal motility). On average, the ratio of *Firmicutes*:*Bacteroidota* among the participants of this study was ~4.0, and age did not have a significant impact on the ratio. However, it should be noted that a decline in microbiological diversity and a decrease in *Bifidobacterium* spp. could also be due to dietary habits, especially low whole-grain intake [6].

The *Firmicutes*:*Bacteroidota* ratio is frequently cited in the scientific literature also as an indicator of excess weight, yet the explanation for it is not clear [37,38]. Individuals with a *Firmicutes*:*Bacteroidota* ratio of ≥ 1 are 23% more likely to be overweight than those with a *Firmicutes*:*Bacteroidota* ratio of <1 [39]. For all participants, the *Firmicutes*:*Bacteroidota* ratio were ≥1 and no significant association in this study was found regarding body mass index category and *Firmicutes*:*Bacteroidota* ratio. Also, other parameters related to the evaluation of body composition, like waist circumference and waist-to-hip ratio were not associated with differences in *Firmicutes: Bacteroidota* ratio.

Some studies [9] have reported that oestrogen deficiency and postmenopausal period compared to premenopausal period are associated with changes in the gut microbiome (lower abundance of phylum *Firmicutes* and genus *Roseburia*, higher abundance of phylum *Bacteroidota* and genus *Tolumonas)*. In this study, only postmenopausal women (no menstrual bleeding for at least 12 months) were participating, therefore, it is not possible to evaluate if similar observations of gut microbiome changes also apply to postmenopausal women in Latvia.

Few studies have also indicated gut microbiome composition differences related to bone mineral density, for example, a higher abundance of *Bacteroides* spp. among women with osteopenia or osteoporosis [11,12]. Based on the DEXA scan performed before the study, 23 participants had a normal bone mineral density level, 19 participants had osteopenia, but 10 participants had osteoporosis. Yet, no significant changes were observed between bone mineral density and the composition of the gut microbiome.

Cigarette smoke is a source of pollutants (nicotine, polycyclic aromatic hydrocarbons, heavy metals, etc.) and can influence the composition of the human gut microbiome [40]. Smoking is associated with an increased abundance of the phylum *Bacteroidota* and with decreased abundance of phylum *Firmicutes* and phylum *Proteobacteria* compared to persons who have never smoked [41]. Yet, the mechanism for it is not clear, because the majority of studies analysing the effect of smoking-related pollutants on the gut microbiome are performed on animals, not humans [40]. The majority of the participants of this study were non-smokers and no significant impact of smoking on the gut microbiome was observed.

Most participants noted the use of alcohol two to four times a month or more often, and no significant associations were found regarding alcohol intake frequency and composition of the gut microbiome.

Gut microbiome composition can be affected by the use of antibiotics leading to changes in the gut microbiome. For example, the use of macrolide class antibiotics like clarithromycin leads to a decreased abundance of phylum *Firmicutes* and an increased abundance of phyla *Bacteroidota* and *Proteobacteria* [7]. Nevertheless, the alteration of the gut microbiome depends not only on the antibiotic class but also on the dose, period of exposure, and other factors [7]. Only a few participants from this study (six participants from the control group and four participants from the experimental group) noted the use of antibiotics in the recent year, and it was not associated with the differences within the gut microbiome.

Dietary habits are one of the key modulators of the gut microbiome [42]. Some researchers [43] have reported that higher dietary quality (more precisely, frequent whole grain and vegetable intake) is associated with a greater gut microbiome diversity (a higher alpha-diversity index). Yet, reported associations from this study [43] were only weakly positive (ρ = 0.265, *p* = 0.015 for vegetable intake and ρ = 0.264, *p* = 0.015 for whole grains intake).

In this study, specific food product and beverage intake were not associated with a higher alpha-diversity index. However, we observed that grain and grain-based product intake was associated with a higher abundance of *Bifidobacterium* spp. Many positive health aspects have been associated with the consumption of whole-grains and abundance of *Bifidobacterium* spp. [44,45,46]. For example, *Bifidobacterium* spp. can help to prevent gastrointestinal infections by competitively excluding pathogens from common binding sites on gut epithelial cells [44]. Grains and grain-based products, and more precisely, whole grains, are also a source of non-digestible fibre that through their metabolisation by microorganisms in the gut, modulate the composition and/or activity of the gut microbiome, thus providing a beneficial effect for the host [44]. Therefore, it is important to include grains and grain-based products in the daily diet as they contribute to a healthy gut microbiome [45]. Yet, grains and grain-based product intake among study participants was low. A previous study conducted in Latvia evaluating dietary habits among the Latvian adult population [47] observed low grains and grain-based product intake (~175 g per day) not only among postmenopausal women (50 to 64 years old) but also women aged 19–34 years (~190 g per day) and women aged 35–49 (~208 g per day), indicating that low grains and grain-based product intake is common among women in Latvia regardless of age [47].

The animal-based diet increases the abundance of genus *Alistipes*, *Bilophila*, and *Bacteroides*, but a plant-based diet increases the abundance of microorganisms from phylum *Firmicutes* that metabolise dietary plant polysaccharides (*Roseburia* spp., *Eubacterium rectale*, and *Ruminococcus bromii*) [7,39]. However, no significant correlation was established between animal-based product intake and abundance of microorganisms from the genus *Alistipes*, *Bilophila*, and *Bacteroides* in this study, and on the contrary, a significant negative association was found between fruit and berry intake and abundance of genus *Ruminococcus*.

The abundance of *Subdoligranulum* spp. was associated with milk and milk product intake among the participants of this study. Recent experimental studies [48] have reported that the abundance of *Subdoligranulum* spp. is positively associated with a higher high-density lipoprotein cholesterol level but negatively associated with fat mass, adipocyte diameter, insulin resistance, levels of leptin and insulin, therefore could be associated with better health. However, more studies are required to evaluate observed associations [48].

The yoghurt bacteria *Streptococcus thermophilus* and *Lactobacillus bulgaricus* survive the gastrointestinal transit, but generally reach low fecal concentrations (10^4^ to 10^6^ CFU g^−1^ feces) in comparison with resident gut microbes [18]. This could potentially explain why no significant changes in the gut microbiome were found after eight weeks of daily consumption of plain organic milk yoghurt among participants from the experimental group. Obtained results of this study also coincide with data reported by Le Roy et al., 2022 [18], indicating that the changes in the gut microbiome following yoghurt consumption might be transient and that yoghurt bacteria are outcompeted by resident gut bacteria.

Nevertheless, fermented milk product intake is still associated with a positive effect on the gut microbiome (higher alpha-diversity, higher abundance of phylum *Firmicutes*, etc.) [18]. Other positive effects have also been associated with yoghurt consumption. Live yoghurt cultures (*Lactobacillus bulgaricus* and *Streptococcus thermophilus*) in yoghurt improve digestion of lactose in individuals with lactose maldigestion [49,50]. Consumption of yoghurt is also associated with a lower risk for hypertension and lower serum cholesterol and triglycerides levels [50]. Due to high nutrient (protein and calcium) content, yoghurt consumption may also have a beneficial effect on bone health, which is especially important for women during the postmenopausal period [50].

Therefore, the consumption of fermented milk products, including yoghurt, should be recommended on a daily basis. A healthy gut microbiome is important for optimal metabolic and immune functions and the prevention of diseases among postmenopausal women [18].

## 5. Conclusions

This is the first study reporting the gut microbiome composition among postmenopausal women in Latvia. Although no significant changes in the composition of the gut microbiome were related to daily consumption of plain organic milk yoghurt, we report that consumption of food products like grains, grain-based products, milk and milk products, etc. and beverages (tea, coffee) are associated with differences in the composition of the gut microbiome. Nevertheless, very little is known about most of the representatives of gut microbiome, making it difficult to analyse the obtained data. Establishing nutritional strategies aiming to shape the gut microbiome could contribute to improved health status in postmenopausal women, therefore, further research should be continued. Possible modifications of plain organic milk yoghurt microflora by adding other probiotics belonging to *Lactobacillus* spp. and *Bifidobacterium* spp. in addition to *Lactobacillus bulgaricus* and *Streptococcus thermophilus*, could potentially lead to beneficial changes in the composition of the gut microbiome among postmenopausal women but further research is needed.

Future studies should also make use of analytical tools that provide higher resolution for microbiome investigation (such as shotgun metagenomics) and capture the various processes taking place in the host organism and its microbiomes (like metabolomics and transcriptomics). Extended longitudinal studies with multiple sampling points would also allow a better evaluation of the long-term effects of nutritional interventions.

## Figures and Tables

**Figure 1 nutrients-14-03568-f001:**
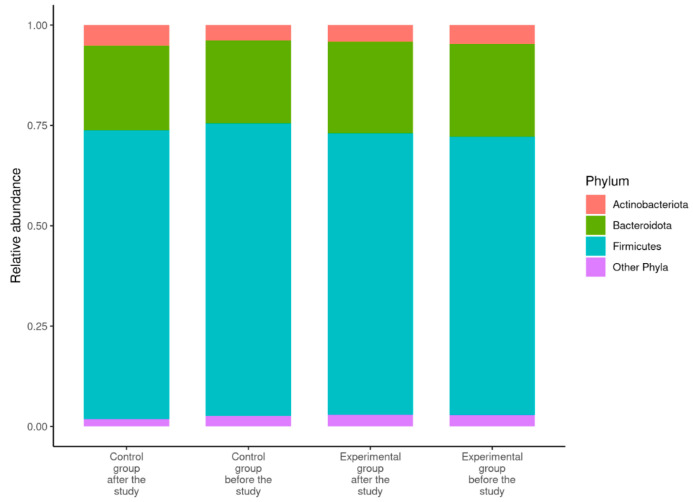
The average gut microbiome composition at the phylum level among study participants (*n* = 52). Phyla with relative abundance below 2% were collapsed in order to avoid cluttering the graph.

**Figure 2 nutrients-14-03568-f002:**
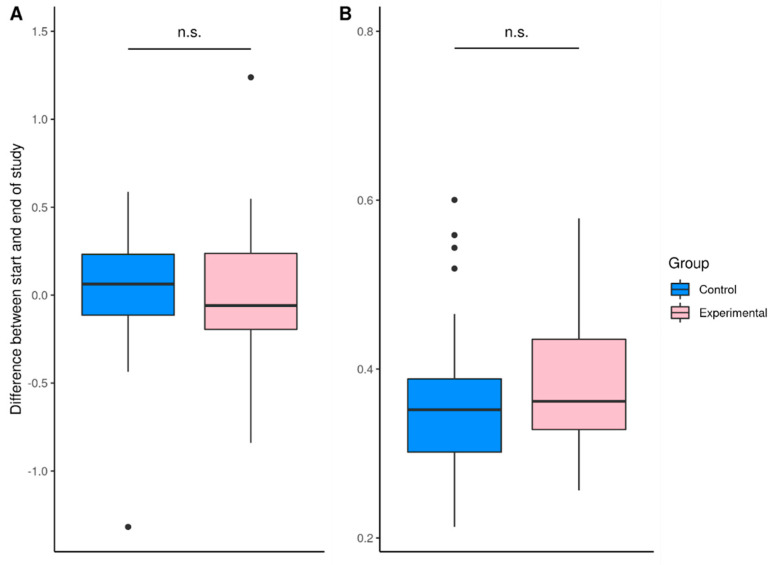
Comparison of the shift in alpha diversity (**A**) and beta diversity (**B**) between the start and end of the study. n.s.: not significant.

**Figure 3 nutrients-14-03568-f003:**
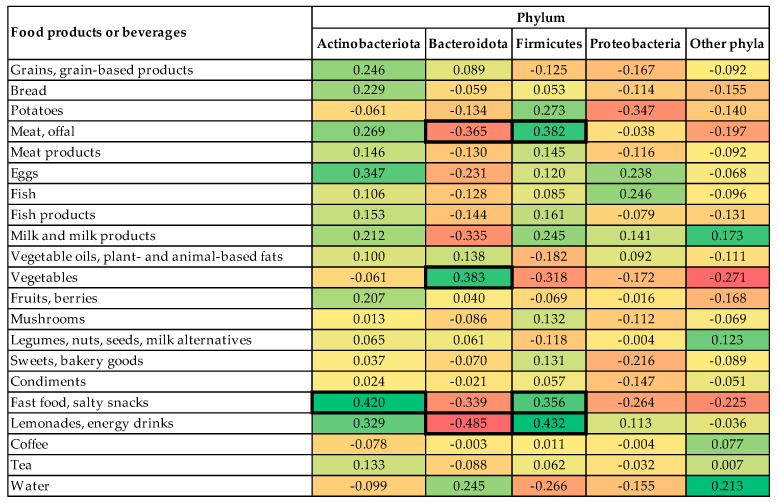
Spearman’s correlations between the abundances of gut microbiome at the phylum level and food and beverage intake among the study participants. Red: negative correlations; yellow: no correlation; green: positive correlations. *p*-value < 0.05 for bolded cells.

**Table 1 nutrients-14-03568-t001:** Description of the yoghurt consumed among the participants from the experimental group.

Packaging: Cup Weight/Capacity: 350 g List of Ingredients: Milk, Milk Protein, Lactic Acid Bacteria 1.4 × 10^9^ CFU g^−1^	
Nutrition Declaration	Values Per 100 g
Energy value	289–327 kJ/69–78 kcal
Fat, of which	3.5–4.5 g
–saturates	2.4–3.0 g
Carbohydrates, of which	4.5 g
–sugars	4.5 g
Protein	4.8 g
Salt ^1^	0.10 g
Calcium	132 mg
Vitamin D	<0.10 μg

^1^ the salt content is exclusively due to the presence of naturally occurring sodium.

**Table 2 nutrients-14-03568-t002:** Characteristics of the participants.

Characteristics	Control Group (*n* = 26)	Experimental Groups (*n* = 26)
Age (years)	58 ± 5 (49–69)	57 ± 4 (51–69)
Education level	Secondary education (*n* = 2) Higher education (*n* = 24)	Secondary education (*n* = 2) Higher education (*n* = 24)
Body mass index (kg m^−2^)	27.82 ± 4.62 (20.87–38.93)	28.35 ± 6.26 (19.20–44.47)
Waist circumference (cm) ^1^	93 ± 12 (72–112)	93 ± 14 (73–124)
Waist-to-hip ratio ^1^	0.86 ± 0.08 (0.70–1.01)	0.85 ± 0.07 (0.76–1.02)
Last menstrual period ^1^	12 months (*n* = 4) Before 13–24 months (*n* = 1) More than 24 months ago (*n* = 20)	12 months (*n* = 2) Before 13–24 months (*n* = 7) More than 24 months ago (*n* = 17)
Smoking	Never (*n* = 15) Used to smoke (*n* = 8) Occasionally (*n* = 2)	Never (*n* = 20) Used to smoke (*n* = 4) Occasionally (*n* = 2)
	Daily (*n* = 1)	Daily (*n* = 0)
Use of alcohol	Never (*n* = 2) Once a month or less often (*n* = 6) 2 to 4 times a month (*n* = 14) 2 to 3 times a week (*n* = 2) 4 and more times a week (*n* = 2)	Never (*n* = 2) Once a month or less often (*n* = 11) 2 to 4 times a month (*n* = 10) 2 to 3 times a week (*n* = 3) 4 and more times a week (*n* = 0)
Bone mineral density level	Normal (*n* = 10) Osteopenia (*n* = 8) Osteoporosis (*n* = 8)	Normal (*n* = 13) Osteopenia (*n* = 11) Osteoporosis (*n* = 2)
Use of antibiotics in the last year	Yes (*n* = 6) No (*n* = 20)	Yes (*n* = 4) No (*n* = 22)

^1^ One participant did not provide an answer.

**Table 3 nutrients-14-03568-t003:** Characteristics of the gut microbiome (*n* = 52).

Characteristics	Control Group (*n* = 26)		Experimental Group (*n* = 26)	
	Before the Study	After the Study	Before the Study	After the Study
Ratio of *Firmicutes*:*Bacteroidota*	4.06 ± 1.71 ^1^ (1.78–7.87)	4.06 ± 2.12 (1.41–10.11)	3.65 ± 2.01 (1.27–8.58)	4.07 ± 3.03 (1.03–14.43)
Ratio of *Proteobacteria*:*Actinobacteria*	0.29 ± 0.29 (0.02–1.19)	0.24 ± 0.28 (0.00–1.11)	0.37 ± 0.48 (0.02–2.39)	0.41 ± 0.69 (0.01–2.77)

^1^ Mean ± standard deviation (minimal–maximal value).

## Data Availability

The data that support the findings of this study are available on request from the corresponding author [L.A.]. The data are not publicly available due to containing information that could compromise the privacy of research participants.

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
