# Peer review of "The Gut Microbiome among Postmenopausal Latvian Women in Relation to Dietary Habits"

_nutrients, 2022, doi:10.3390/nu14173568_

Round 1
Reviewer 1 Report
Interesting study. Some comments from me.
Method:
Need information on how each participant is assigned to the respective group.
Is the p-value corrected ?
Authors mentioned in the method that one of the inclusion criteria is postmenopausa which is at least 12 month, but Table 2 show, under the Last menstrual period “before 12 months”? Should it be “Within 12 months” or “12 months” to avoid confusion? Also, instead of just stating those numbers here for each category, has the author performed any statistical test to see if there is any significant difference between them?
Results:
Result Table 3: is it mean+/-std? Need an indication below the table.
Figure 2: how about the alpha and beta diversity based on antibiotic usage?
Has the author analysed individual data in addition to the overall data as a whole ? The microbiome is idiosyncratic with significant inter-individual variation, it might be worth plotting the data for each individual to see if there is a pattern or with different timepoints if data is available.
Discussion:
General: suggest removing all the p-values as those are considered results and thus should be in the result section.
Line 338: Is “waist circumference” and “waist-to-hip ratio” considered a “non-communicable disease”?
Line 357: How about secondhand smoke? Same effect with smoking? does “…never smoke” take into account secodhand smoke?
Line 366-373: This study (as the main aim) should be focusing on yoghurt consumption in postmenopausal women, don’t understand the purpose of this paragraph here.
Line 389-400: How about Bifidobacterium composition in postmenopausal women in previous studies? Or at least in women in general? Or the effect of consuming grain-based products on women?
Line 433: this study found no significant difference between the experimental and control groups. However, the statement here says daily yoghurt intake should be recommended to postmenopausal women?
Conclusion:
Line 442: “…may be associated” sounds ambiguous. We all know that gut microbiome composition changes based on dietary habits. It should replace with stronger conclusion to show the importance of this study.
Any suggestion for future study, yoghurt to postmenopausal women?
Author Response
We would like to thank the reviewer for the detailed comments and suggestions provided for the improvement of the manuscript (nutrients-1784541). We believe that the comments have identified important areas which required improvement. After completion of the suggested edits, the revised manuscript has benefited from an improvement in the overall presentation and clarity. Below, you will find a point-by-point description of how each comment was addressed in the manuscript. Original comments in boldface, responses in regular typeface.
Need information on how each participant is assigned to the respective group.
We added information that participants were randomly divided to the respective group (Line 109)
Is the p-value corrected ?
No
Authors mentioned in the method that one of the inclusion criteria is postmenopausa which is at least 12 month, but Table 2 show, under the Last menstrual period “before 12 months”? Should it be “Within 12 months” or “12 months” to avoid confusion? Also, instead of just stating those numbers here for each category, has the author performed any statistical test to see if there is any significant difference between them?
We replaced word “before 12 months” with “12 months” in Table 2. Due to small sample size we did not performed any statistical test for these groups.
Results:
Result Table 3: is it mean+/-std? Need an indication below the table.
We added clarification “Mean ± standard deviation (minimal–maximal value)” below the Table 3.
Figure 2: how about the alpha and beta diversity based on antibiotic usage?
As antibiotic usage in the last year was reported only by small number of participants (n=10), we did not evaluated it further.
Has the author analysed individual data in addition to the overall data as a whole ? The microbiome is idiosyncratic with significant inter-individual variation, it might be worth plotting the data for each individual to see if there is a pattern or with different timepoints if data is available.
We added some additional statistical Spearman correlation coefficient data (Figure 6) and data from principal component analysis (Table 7). Unfortunately, overall, we faced difficulties to analyse the obtained data as the most of the performed statistical analysis did not lead to significant results.
Discussion:
General: suggest removing all the p-values as those are considered results and thus should be in the result section.
We removed all p-values form the Discussion section, except for statistical data reported as information from the different study (Line 368–369).
Line 338: Is “waist circumference” and “waist-to-hip ratio” considered a “non-communicable disease”?
We rewrite this sentence to make it clearer – “Also, other parameters related to the evaluation of body composition, like waist circumference and waist-to-hip ratio were not associated with differences in Firmicutes:Bacteroidota ratio” (Lines 327–329).
Line 357: How about secondhand smoke? Same effect with smoking? does “…never smoke” take into account secodhand smoke?
We did not take secondhand smoke into consideration as it is difficult to evaluate all sources of it:
- smoking relative in home;
- smoking neighbours and smoke coming into home via window or ventilation systems;
- smoking colleague in work;
Line 366-373: This study (as the main aim) should be focusing on yoghurt consumption in postmenopausal women, don’t understand the purpose of this paragraph here.
We removed it.
Line 389-400: How about Bifidobacterium composition in postmenopausal women in previous studies? Or at least in women in general? Or the effect of consuming grain-based products on women?
We added additional source and information that aging is related to decrease in abundance of Bidifobacterium (Lines 315–316). Also, more information regarding consumption of whole-grains and higher abundance of Bifidobacterium were provided (Lines 371–374).
Line 433: this study found no significant difference between the experimental and control groups. However, the statement here says daily yoghurt intake should be recommended to postmenopausal women?
Although, no significant difference was found regarding yoghurt consumption and gut microbiome in this study, we cannot make a statement that yoghurt should not be consumed by postmenopausal women. Yoghurt is still a good source of calcium, easily digested proteins, etc. to include in the everyday diet, therefore we included this statement – “Yoghurt consumption still should be recommended for postmenopausal women due to high nutrient content and benefits for the bone health” (Lines 415–418).
Conclusion:
Line 442: “…may be associated” sounds ambiguous. We all know that gut microbiome composition changes based on dietary habits. It should replace with stronger conclusion to show the importance of this study.
We replaced “may be associated” with “is associated”.
Any suggestion for future study, yoghurt to postmenopausal women?
At the end of conclusions we added some suggestions for possible further research regarding yoghurt intake and gut microbiome among postmenopausal women.
Reviewer 2 Report
The article “The Gut Microbiome Among Postmenopausal Latvian Women. The Influence of Dietary Habits.” investigates the role of dietary habits on the microflora of Postmenopausal Latvian Women. The study is interesting and shows that consumption grains, grain-based products, milk and milk products, etc. and beverages may be associated with differences in the composition of the gut microbiome. However, no significant association of microbiota with use of yogurt was found. The gut microbiome analysis presented in the manuscript is of basic level and do not truly represent all aspects of data. The manuscript can be greatly improved by advance level analysis and it will also increase the overall scope of the study
Comments
1. Line 86 explain how the study design is unique and different from the previously reported studies
2. Line # 95 Why was recent intake of antibiotics not considered as an exclusion criterion as it may have a drastic effect the gut microbiota?
3. Line # 95 were women who had hysterectomy also recruited in the study or were excluded
4. Line # 132-174 Please represent the data in a form of table
5. Line 175 why was microbiome analysis done after 8 weeks of yogurt consumption. Is this duration enough to result in significant change in microbiota please provide reference?
6. Line # 254-284 make a comparative table to indicate genus significantly associate with the groups and also elaborate them in discussion section
7. Over all the microbiome analysis is of preliminary level and the article may benefit from a more comprehensive analysis.
Author Response
We would like to thank the reviewer for the detailed comments and suggestions provided for the improvement of the manuscript (nutrients-1784541). We believe that the comments have identified important areas which required improvement. After completion of the suggested edits, the revised manuscript has benefited from an improvement in the overall presentation and clarity. Below, you will find a point-by-point description of how each comment was addressed in the manuscript. Original comments in boldface, responses in regular typeface.
- Line 86 explain how the study design is unique and different from the previously reported studies
Explanation was added (Lines 87–91).
- Line # 95 Why was recent intake of antibiotics not considered as an exclusion criterion as it may have a drastic effect the gut microbiota?
Starting the study, antibiotic usage was not determined as the exclusion criteria for the participants. However, information regarding the usage of antibiotics was collected and data used while analysing the obtained results. Only small number of participants noted the use of antibiotics (n=10). In the end, according to the results of this study, no significant differences in the gut microbiome were related to the antibiotic use.
- Line # 95 were women who had hysterectomy also recruited in the study or were excluded
As one of the inclusion criteria was “no menstrual bleeding for at least 12 months”, women who had hysterectomy were automatically excluded from the study.
- Line # 132-174 Please represent the data in a form of table
We moved data to the Table 2.
- Line 175 why was microbiome analysis done after 8 weeks of yogurt consumption. Is this duration enough to result in significant change in microbiota please provide reference?
We added explanation why 8 week period was selected (Lines 111–112).
- Line # 254-284 make a comparative table to indicate genus significantly associate with the groups and also elaborate them in discussion section
Data regarding significant correlations between specific product intake and gut microbiome were moved to the Table 6. Also, additional Figure (No. 6) were added to the manuscript evaluating correlation between food, beverage intake and abundance of specific gut microorganisms at the phylum level.
- Over all the microbiome analysis is of preliminary level and the article may benefit from a more comprehensive analysis.
We added some additional statistical Spearman correlation coefficient data (Figure 6) and data from principal component analysis (Table 7). Unfortunately, overall, we faced difficulties to analyse the obtained data as the most of the performed statistical analysis did not lead to significant results.
Round 2
Reviewer 1 Report
Thanks for the revised version.
A few comments from me.
Table 7 is redundant, can the author plot it in figure form?
Line 315-320: Authors mention that a decrease in Bifidobacterium spp might be related to the elderly digestive system. Does the author think that it has any relationship with food? For example, breast-fed babies have higher Bifidobacterium when compared to their formula-fed counterparts.
Line 370- 381: ".......... Yet, grains and grain-based product intake among study participants was 381
low".
So, does this low grain-based product intake relate to the duration of postmenopausal among the participants?
Conclusion:
".. adding probiotics belonging to Lactoba- 429
cillus spp. and Bifidobacterium spp. could led to differences in the composition of the gut 430
microbiome among postmenopausal women".
Lactobacillus spp. and Bifidobacteria are already common in yoghurt, so this suggestion seems redundant?
Overall there seems to have too many tables, might be good to move some to supplementary if ever?
Author Response
We thank the Reviewer for the second evaluation. Below, you will find a point-by-point description of how each comment was addressed in the manuscript. Original comments in boldface, responses in regular typeface.
Table 7 is redundant, can the author plot it in figure form?
We tried to plot data in figure (bi-plot), however it did not make the data clearer. Therefore, we decided to move this Table together with other “big” Tables to the Supplementary materials.
Line 315-320: Authors mention that a decrease in Bifidobacterium spp might be related to the elderly digestive system. Does the author think that it has any relationship with food? For example, breast-fed babies have higher Bifidobacterium when compared to their formula-fed counterparts.
Yes, it could be also related to dietary habits. We added additional explanation (Lines now 308 to 310) --> However, it should be noted that a decline in microbiological diversity and a decrease in Bifidobacterium spp. could be also due to dietary habits, especially low whole-grain intake [6].
Line 370- 381: ".......... Yet, grains and grain-based product intake among study participants was low". So, does this low grain-based product intake relate to the duration of postmenopausal among the participants?
Unfortunately, low grains, grain-based product intake is low among women in Latvia regardless of age. We added this information in the manuscript --> Yet, grains and grain-based product intake among study participants was low. Previous study conducted in Latvia evaluating dietary habits among Latvian adult population [48] has observed low grains and grain-based product intake (~175 g per day) not only among postmenopausal women (50 to 64 years old) but also women aged 19–34 years (~190 g per day) and women aged 35–49 (~208 g per day), indicating that low grains and grain-based product intake is common among women in Latvia regardless of age [48] (Lines now 370 to 375).
Conclusion:
".. adding probiotics belonging to Lactobacillus spp. and Bifidobacterium spp. could led to differences in the composition of the gut microbiome among postmenopausal women".
Lactobacillus spp. and Bifidobacteria are already common in yoghurt, so this suggestion seems redundant?
Yoghurt used during this study containing only Lactobacillus bulgaricus and Streptococcus thermophilus did not lead to differences of gut microbiome but additional probiotics in yoghurt may affect gut microbiome. Therefore, we rephrased this sentence to make this idea clearer ---> “Possible modifications of plain organic yoghurt microflora by adding other probiotics belonging to Lactobacillus spp. and Bifidobacterium spp. in addition to Lactobacillus bulgaricus and Streptococcus thermophilus could potentially affect the composition of the gut microbiome among postmenopausal women but further research is needed.”. (Lines now 421 to 425).
Overall, there seems to have too many tables, might be good to move some to supplementary if ever?
We moved the “big” tables – Table 2, Table 4, Table 6, Table 7 to the Supplementary materials.
Reviewer 2 Report
The authors have addressed all the comments.
Line 87-91 plesse add reference
Author Response
Dear Reviewer,
Thank you for the second review. We added references to the Lines (87-91).